# Progressive Depth Up-scaling via Optimal Transport

## Abstract

Scaling Large Language Models (LLMs) yields performance gains but incurs substantial training costs. Depth up-scaling offers training efficiency by adding new layers to pre-trained models. However, most existing methods copy or average weights from base layers, neglecting neuron permutation differences. This limitation can potentially cause misalignment that harms performance. Inspired by applying Optimal Transport (OT) for neuron alignment, we propose Optimal Transport Depth Up-Scaling (OpT-DeUS). OpT-DeUS aligns and fuses Transformer modules in adjacent base layers via OT for new layer creation, to mitigate neuron permutation mismatch between layers. OpT-DeUS achieves better overall performance and offers improved training efficiency than existing methods for continual pre-training and supervised fine-tuning across different model sizes. To further evaluate the impact of interpolation positions, our extensive analysis shows that inserting new layers closer to the top results in higher training efficiency due to shorter back-propagation time while obtaining additional performance gains. We also find a strong correlation between strong depth up-scaling performance and high transport matrix entropy. Code is provided in the supplementary material.

## 1 Introduction

Large Language Models (LLMs) performance is largely attributed to scaling laws, where capabilities often improve with increased model and data size (Brown et al., 2020; Kaplan et al., 2020; Wei et al., 2022; Chung et al., 2024). However, scaling poses significant sustainability challenges, stemming from increased computational and data demands. Computational demands include hardware constraints (Thompson et al., 2022), carbon emissions (Luccioni et al., 2023; Luccioni & Hernandez-Garcia, 2023) and energy consumption (Wu et al., 2022; de Vries, 2023). Data-related demands involve dataset exhaustion (Villalobos et al., 2024), and quality problems (Luccioni & Viviano, 2021; Bender et al., 2021; Birhane et al., 2023).

To address these challenges, "smart scaling" approaches such as model expansion have been proposed. Model expansion increases the parameter size of a pre-trained model without changing the original architecture. This includes increasing the number of layers, i.e. depth up-scaling (Kim et al., 2024; Wu et al., 2024; Yang et al., 2025; Du et al., 2024), or neurons per layer, i.e. width up-scaling (Samragh et al., 2024). Furthermore, approaches that combine depth and width up-scaling have also been proposed (Shen et al., 2022; Wang et al., 2023; 2024; Yao et al., 2024).

Unlike earlier methods that focus on updating the entire model (Shen et al., 2022; Kim et al., 2024; Du et al., 2024; Wang et al., 2024), recent progressive depth up-scaling approaches update only the newly added layers. This approach enhances training efficiency while mitigating catastrophic forgetting (Kim et al., 2024; Yang et al., 2025). Typically, new layers are initialized by copying (Wu et al., 2024; Kim et al., 2024; Du et al., 2024) or averaging (Yano et al., 2025) from base layers. Copying or averaging from base layers for new layer initialization, while effective, neglects neuron permutation mismatch that can harm downstream performance (Li et al., 2015; Yurochkin et al., 2019a;b). An alternative method (Yang et al., 2025) trains an auxiliary neural network for new layer initialization, but it is sensitive to model layers. These challenges motivate our main research question: *How to effectively initialize new layers to avoid neuron permutation mismatches in progressive depth up-scaling?*

Inspired by applying Optimal Transport (OT) (Singh & Jaggi, 2020; Imfeld et al., 2024), we propose Optimal Transport Depth Up-Scaling (OpT-DeUS) for progressive depth up-scaling. As shown in Figure 1, OpT-DeUS aligns and fuses adjacent layers module-wise to create neuron-aligned new layers. Newly added layers are initialized via OT and inserted into the top half of the base model. Certain module weights are set to zero for better neuron alignment and function preservation. Our contributions are as follows:

- We introduce OpT-DeUS, which creates intermediate layer from adjacent layers by neuron alignment via OT. Experiments show that OpT-DeUS outperforms existing baselines on both continual pre-training and supervised fine-tuning training stages across various model sizes and diverse tasks.

- OpT-DeUS achieves top overall efficiency among baselines. Our comprehensive study on layer interpolation position shows that inserting new layers at higher positions leads to higher training efficiency due to decreased back-propagation time while obtaining better performance.

- OpT-DeUS mitigates neuron permutation mismatch, evidenced by the better performance compared to averaging without neuron alignment. Furthermore, our entropy analysis reveals a correlation between strong performance and high transport matrix entropy when intializing new layers.

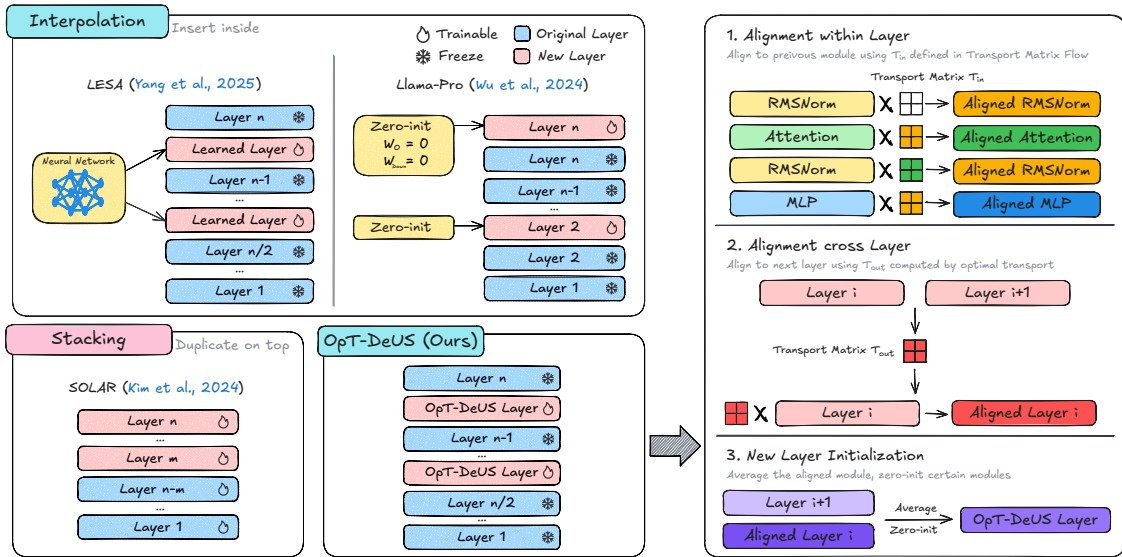

Figure 1: State-of-the-art depth up-scaling methods and our proposed OpT-DeUS. OpT-DeUS uses optimal transport to initialize new layers, each derived from two adjacent base layers $f_i$ and $f_{i+1}$. It first aligns each module $b$ to previous module $b-1$ in $f_i$ (i.e., Alignment within Layer), then aligns it to $b$ in $f_{i+1}$ (i.e., Alignment across Layer). Each colour in OpT-DeUS represents a module, and colour intensity indicates the impact of alignment.

## 2 Related Work

### 2.1 Model Expansion

Model expansion accelerates neural network training by expanding a base pre-trained model to reduce training time and computational overhead (Chen et al., 2016; Wei et al., 2016; Chang et al., 2018; Rusu et al., 2022). Network architecture preservation has proven effective for iterative expansion in encoder-only LLMs (Gong et al., 2019; Yang et al., 2020; Chen et al., 2022). More recently, various model expansion approaches have been explored for decoder-only LLMs. Du et al. (2024) showed depth up-scaling yields greater training efficiency and stronger downstream performance compared to width up-scaling. However,

prior work primarily focuses on expansion during the pre-training stage with a relatively large pre-training corpus (Shen et al., 2022; Wang et al., 2023; 2024; Yao et al., 2024; Yano et al., 2025), resulting in high overall computational costs. Limited work focuses on post-training expansion (Kim et al., 2024; Wu et al., 2024; Yang et al., 2025), using a substantially smaller corpus compared to the original pre-training corpus for training efficiency.

## 2.2 Depth Up-Scaling

**Stacking.** Stacking methods insert a successive of new layers, typically on top of the base model by copying the pre-trained weights of the base model (Du et al., 2024; Kim et al., 2024). Du et al. (2024) proposed stacking entire base layers for stronger downstream performance during pre-training. Kim et al. (2024) introduced SOLAR, a partial stacking approach that omits the copying of the bottom and top layers for new model initialization. SOLAR is effective for continual pre-training. However, stacking requires updating the entire model, incurring extra computational costs.

**Interpolation.** Interpolation methods insert new layers inside the base model. Previous work focuses on creating function preservation layers, where the expanded model performs identically to the base model prior to further training. Achieving function preservation leads to steadier learning processes and better performance. This is achieved by setting the LayerNorm weights to zero for new layer initialization (Shen et al., 2022), initializing the entire new layer to zero (Wang et al., 2024), or employing dynamic masking mechanisms (Yao et al., 2024). Wu et al. (2024) proposed LLaMA PRO, which initializes the inserted new layers by copying weights from the base model. For function preservation, the output matrices of attention and MLP in these new Transformer layers are set to zero, termed zero-initialization. Yano et al. (2025) initialized new layers by averaging weights from adjacent base layers for pre-training. They fully updated the new layers while applying a parameter-efficient fine-tuning approach to the base layers. LESA (Yang et al., 2025) initializes new layers using an auxiliary network given adjacent layers at interpolation positions as input. However, existing methods largely rely on copying (Kim et al., 2024; Wu et al., 2024) or averaging (Yano et al., 2025) to initialize new layers, neglecting neuron permutation differences.

## 2.3 Progressive Depth Up-Scaling

Progressive depth up-scaling, exemplified by LLaMA PRO and LESA, enables knowledge injection while mitigating catastrophic forgetting by only updating the inserted new layers. Recent work has used progressive depth up-scaling for language adaptation (Choudhury et al., 2025; Hennara et al., 2025). It preserves the parametric knowledge of base layers while allowing new knowledge to be learned in the expanded layers. However, while existing methods use different strategies to expand the layers of the model, little focus has been placed on the impact of interpolation positions regarding training efficiency.

# 3 Preliminaries

## 3.1 Depth Up-scaling

Let $\mathcal{M}$ be a *base* LLM with $n$ Transformer layers $\{f_i\}_{i=1}^n$, parametrized by $\theta$. The aim is to obtain an *expanded* model $\mathcal{M}'$ with parameters $\theta'$ by introducing $k$ additional Transformer layers $\{f'_i\}_{i=1}^k$. $\mathcal{M}'$ retains the same layer type (i.e. Transformer layers) and hidden dimension of the base model.

Each Transformer layer $f_i$ is composed of a sequence of modules. We denote the parameters of layer $f_i$ by $\{\mathbf{W}_b^{(i)}\}_{b=1}^B$, where $\mathbf{W}_b^{(i)}$ represents the weight matrix of module $b$ in layer $f_i$. Accordingly, each new layer $f'_i$ is parameterized by $\{\mathbf{W}'_b^{(i)}\}_{b=1}^B$.

**Stacking.** $\mathcal{M}$ is expanded by adding a set of new layers on top of the base layers to obtain $\mathcal{M}'$. $\circ$ denotes the connection between Transformer layers:

$$\mathcal{M}'(x;\theta') = f'_k \circ \cdots \circ f'_1 \circ f_n \circ \cdots \circ f_1(x).$$

Each new layer $f_i'$ is typically initialized by duplicating the parameters of a base layer $f_i$ (Du et al., 2024; Kim et al., 2024). Concretely, this corresponds to module-wise weight copying:

$$\mathbf{W'}_b^{(i)} \leftarrow \mathbf{W}_b^{(i)}, \quad \forall b \in \{1, \ldots, B\}.$$

**Interpolation.** Figure 1 illustrates different interpolation strategies adopted by existing depth up-scaling methods. $\mathcal{M}$ is expanded by inserting new layers between base layers as follows:

$$\mathcal{M}'(x; \theta') = \begin{cases} f_i' \circ f_i, & \text{if a new layer is inserted,} \\ f_i, & \text{otherwise.} \end{cases}$$

Unlike stacking, interpolation methods initialize each new layer $f_i'$ from two adjacent base layers $f_i$ and $f_{i+1}$. At the parameter level, this corresponds to initializing each module weight matrix $\mathbf{W'}_b^{(i)}$ using the corresponding modules in $f_i$ and $f_{i+1}$. Existing approaches initalized $f_i'$ by copying (Wu et al., 2024; Kim et al., 2024; Du et al., 2024), averaging (Yano et al., 2025), or prediction via an auxiliary network (Yang et al., 2025). Formally, for $\forall b \in \{1, \ldots, B\}$ :

$$\mathbf{W'}_b^{(i)} \leftarrow \begin{cases} \mathbf{W}_b^{(i)}, & \text{copying,} \\ \mathrm{Avg}\left(\mathbf{W}_b^{(i)}, \mathbf{W}_b^{(i+1)}\right), & \text{averaging,} \\ \mathrm{NN}\left(\mathbf{W}_b^{(i)}, \mathbf{W}_b^{(i+1)}\right), & \text{prediction,} \\ \mathrm{OpT\text{-}DeUS}\left(\mathbf{W}_b^{(i)}, \mathbf{W}_b^{(i+1)}\right), & \text{our method,} \end{cases}$$

## 3.2 Optimal Transport

Optimal Transport is a mathematical framework determining the most cost-effective way to transform one probability distribution into another, given a defined transport cost. Formally, let $\mu = \sum_{i=1}^{n} \alpha_i \delta(x^{(i)})$ and $\nu = \sum_{j=1}^{m} \beta_j \delta(y^{(j)})$ be two discrete probability measures supported on $\{x^{(i)}\}_{i=1}^{n}$ and $\{y^{(j)}\}_{j=1}^{m}$ with probability distribution $\boldsymbol{\alpha} = (\alpha_1, \ldots, \alpha_n)$ and $\boldsymbol{\beta} = (\beta_1, \ldots, \beta_m)$, respectively, where $\delta(s)$ denotes the unit mass at point $s$. Given a cost matrix $\mathbf{C} \in \mathbb{R}^{n \times m}$ where $\mathbf{C}_{ij}$ is the cost of transporting mass from $x^{(i)}$ to $y^{(j)}$, the Optimal Transport (OT) problem is defined as:

$$\mathrm{OT}(\mu, \nu, \mathbf{C}) = \min_{\mathbf{T} \in \mathbb{R}_+^{n \times m}} \sum_{i,j} \mathbf{T}_{ij} \mathbf{C}_{ij} \quad \text{s.t.} \quad \mathbf{T} \mathbf{1}_m = \boldsymbol{\alpha}, \quad \mathbf{T}^\top \mathbf{1}_n = \boldsymbol{\beta}.$$

The transport matrix $\mathbf{T}$ can be obtained via the Earth-Mover's Distance (EMD) (Rubner et al., 2000) for a sparse solution, or the Sinkhorn-Knopp algorithm (Knight, 2008) for a dense solution. Unlike EMD, which typically yields a sparse transport plan corresponding to a near one-to-one (hard) alignment, the Sinkhorn-Knopp algorithm introduces an entropic regularization that encourages smoother transport plans. As a result, probability mass can be softly distributed across multiple target points, leading to a dense transport matrix that can be interpreted as a soft alignment between the two distributions.

# 4 Optimal Transport Depth Up-Scaling

## 4.1 Motivation: Neuron Permutation Mismatch

Copying and averaging weights from original layers are commonly used methods for creating new layers in depth up-scaling (Wu et al., 2024; Kim et al., 2024; Du et al., 2024; Yano et al., 2025). However, this approach neglects the problem of neuron permutation mismatch, which is widely present in deep neural

networks and Transformers (Li et al., 2015; Yurochkin et al., 2019a;b). During training, a single neuron may contribute to multiple functions (Nguyen et al., 2016), and same-index neurons in different layers may not be functionally corresponding (Sajjad et al., 2022; Klabunde et al., 2025). As shown in Figure 2, averaging from base layers to initialize $f_i'$ can incorrectly merge neurons with different functionalities, while copying from $f_i$ to initialize $f_i'$ breaks the neuron connectivity between $f_i$ and $f_i'$. Thus, directly copying or averaging weights can cause misalignment between $f_i$ and $f_{i+1}$, potentially harming performance (Li et al., 2015; Yurochkin et al., 2019a;b).

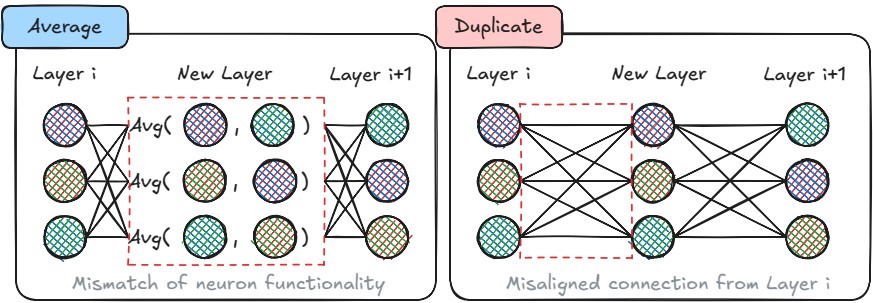

Figure 2: Illustration of neuron permutation mismatch caused by average and duplicate. Each column of neurons represents the order of neurons within layer. Multiple colours of each neuron represent multiple functions it contributes to. Direct averaging weights for new layer $f_i'$ align neurons with mismatched functionality. Duplicating $f_i$ for initializing $f_i'$ can preserve the neuron connection between $f_i'$ and $f_{i+1}$, while the connection between $f_i$ and $f_i'$ is misaligned.

Neuron permutation mismatch can be mitigated by aligning neurons between $f_i$ and $f_{i+1}$ using OT, which models functional similarity per neuron across layers. Singh & Jaggi (2020) and Imfeld et al. (2024) showed that aligning neurons layer-wise via OT leads to better-initialized new layers $f'$ from base layers $f$ for model merging, a shared operation with depth-up scaling. Recent research further shows that using information from adjacent layers provides stronger initialization than random initialization in depth up-scaling (Du et al., 2024; Yano et al., 2025; Yang et al., 2025). This inspires proposing Optimal Transport Depth Up-scaling (OpT-DeUS), illustrated in Figure 1. OpT-DeUS is a progressive interpolation method that updates only $f'$ for training efficiency. It aligns and fuses layers $f_i$ and $f_{i+1}$ module by module (e.g. the query module in the attention component) to create $f_i'$ via OT. OpT-DeUS inserts new layers $f_i'$ in the top half of $\mathcal{M}$, between base layers $f_i$ and $f_{i+1}$. This layer interpolation strategy provides better performance (Section 7.1) and training efficiency (Section 7.2).

## 4.2 Transport Matrix Flow for OpT-DeUS

OpT-DeUS relies on two types of transport matrices: $\mathbf{T_{in}}$ and $\mathbf{T}_{\text{out}}$. Each module weight matrix $\mathbf{W}_b^{(i)}$ in $f_i'$ is assigned a $\mathbf{T}_{\text{in}}$. $\mathbf{T}_{\text{in}}$ aligns $\mathbf{W}_b^{(i)}$ to $\mathbf{W}_{b-1}^{(i)}$ within the layer. $\mathbf{T}_{\text{out}}$ aligns $\mathbf{W}_b^{(i)}$ to $\mathbf{W}_b^{(i+1)}$ across layers. $\mathbf{T}_{\text{in}}$ for $\mathbf{W}_b^{(i)}$ is initialized by reusing the $\mathbf{T}_{\text{out}}$ from the previous module $\mathbf{W}_{b-1}^{(i)}$. $\mathbf{T}_{\text{out}}$ is computed by solving an OT problem (Section 4.3).

| Transport Matrix | Normalization | | Attention | | | | MLP | | |
| :---: | :---: | :---: | :---: | :---: | :---: | :---: | :---: | :---: | :---: |
| | Pre-Attn | Pre-MLP | Query | Key | Value | Output | Gate | Up | Down |
| $\mathbf{T}_{\text{in}}$ | $\mathbf{I}$ | $\frac{1}{2}(\mathbf{T}_O+\mathbf{I})$ | $\mathbf{I}$ | $\mathbf{I}$ | $\mathbf{I}$ | $\underline{\mathbf{I}}$ | $\mathbf{T}_O$ | $\mathbf{T}_O$ | $\underline{\mathbf{I}}$ |
| $\mathbf{T}_{\text{out}}$ | $\mathbf{I}$ | $\frac{1}{2}(\mathbf{T}_O+\mathbf{I})$ | $\mathbf{T}_Q$ | $\mathbf{T}_K$ | $\mathbf{T}_V$ | $\mathbf{T}_O$ | $\mathbf{T}_{\text{Gate}}$ | $\mathbf{T}_{\text{Up}}$ | $\mathbf{T}_{\text{Down}}$ |

Table 1: Transport Matrix Flow. We manually set $\mathbf{T}_{\text{in}}$ to each module for alignment within layer. $\mathbf{T}_{\text{out}}$ is calculated through OT for alignment across layers (except Normalization component). Notably, modules highlighted in underline deviate from the vanilla design (Imfeld et al., 2024).

We use Transport Matrix Flow (TMF) to define the assignment of $\mathbf{T}_{\text{in}}$ for each module in the Attention and MLP components of a Transformer layer (Table 1). Following Imfeld et al. (2024), at the layer entrance of

$f'_i$, $\mathbf{T}_{\text{in}}$ is initialized as the identity matrix $\mathbf{I}$. For residual connections (i.e., Pre-MLP Normalization), $\mathbf{T}_{\text{in}}$ is set by averaging the $\mathbf{T}_{\text{out}}$ from both residual paths (i.e. the layer entrance and the attention output).

However, for the following reasons, vanilla TMF is no longer applicable in our setting, and we therefore introduce the corresponding modifications, theses modifications are further ablated in Section 7.3.

**Architectural Advancements**  Group-Query Attention (Ainslie et al., 2023) and SwiGLU MLP (Shazeer, 2020) introduce dimensional mismatches that makes vanilla TMF inapplicable. As a result, we set $\mathrm{T}_{\text{in}} = \mathbf{I}$ for the Attention Output and MLP Down modules.

**Non-consecutive Alignment**  In our setting, alignment are applied only to non-consecutive $f'_i$. $\mathbf{T}_{in}$ at layer entrance does not carry meaningful cross-layer information. Incorporating this will dilute the effective of alignment. Consequently, for MLP Gate and Up that after the residual connection, we set $\mathbf{T}_{in} = \mathbf{T}_O$.

### 4.3 Weight Initialization with OT

Given the parameters of layers $f_i$ and $f_{i+1}$ and the pre-defined TMF, Algorithm 1 demonstrates how to initialize the new layer $f'_i$ via OT. The overall procedure consists of five steps, described below.

---

**Algorithm 1** Optimal Transport Depth Up-Scaling

---

**Input:** Weight matrices for adjacent base layers $\mathbf{W}_b^{(i)}$, $\mathbf{W}_b^{(i+1)}$; Pre-defined $\mathcal{TMF} = \{\mathbf{T}_{\text{in}}^{(b)}\}_{b=1}^{B}$ (Table 1)
**Output:** Weight matrices for new layers $\mathbf{W'}_b^{(i)}$
1: **for** base layer $f_i$ ($\frac{n}{2} \le i < n$) **do**
2:     **for** each module $b$ **do**
3:         Define $\mu$ and $\nu$ over $\mathbf{W}_b^{(i)}$ and $\mathbf{W}_b^{(i+1)}$, with uniformly distributed $\boldsymbol{\alpha}$ and $\boldsymbol{\beta}$         ▷ Instantiate OT problem
4:         Calculate cost $\mathbf{C}_{kj} = \|\delta(x^{(k)}) - \delta(y^{(j)})\|_2$         ▷ Instantiate OT problem
5:         $\mathbf{T}_{\text{in}} \leftarrow \mathcal{TMF}[b]$         ▷ Retrieve $\mathbf{T}_{\text{in}}$ from $\mathcal{TMF}$
6:         $\mathbf{W}_b^{(i)} \leftarrow \mathbf{W}_b^{(i)} \cdot \mathbf{T}_{\text{in}}$         ▷ Alignment within layer
7:         $\mathbf{T}_{\text{out}} = \text{OT}(\mu, \nu, \mathbf{C})$         ▷ Solve instantiated OT via Sinkhorn–Knopp algorithm
8:         $\mathbf{W}_b^{(i)} \leftarrow \mathbf{T}_{\text{out}}^{\top} \cdot \mathbf{W}_b^{(i)}$         ▷ Align across layer
9:         $\mathbf{W'}_b^{(i)} \leftarrow \frac{1}{2}\big(\mathbf{W}_b^{(i)} + \mathbf{W}_b^{(i+1)}\big)$         ▷ Compute $\mathbf{W'}_b^{(i)}$
10:         **end for**
11:     $\mathbf{W'}_O^{(i)}, \mathbf{W'}_{\text{Down}}^{(i)} \leftarrow \mathbf{0}$         ▷ Zero-initialization
12: **end for**

---

**Step-1: OT problem instantiation.** Based on the definition of OT, we first instantiate $\mu$ and $\nu$ over $\mathbf{W}_b^{(i)}$ and $\mathbf{W}_b^{(i+1)}$, respectively. We initialize their associated probability distribution $\boldsymbol{\alpha}$ and $\boldsymbol{\beta}$ uniformly, treating each neuron equally (cf. line 3). For measuring the difference between neurons, we adopt the weight-based support function $\delta$ (Singh & Jaggi, 2020), where each neuron is represented directly by its weight value, avoiding auxiliary constraints. The transport cost $\mathbf{C}_{kj}$ is then defined as the Euclidean distance between the weight value of the $k$-th neuron in $\mathbf{W}_b^{(i)}$ and the $j$-th neuron in $\mathbf{W}_b^{(i+1)}$ (cf. line 4).

**Step-2: Alignment within layer** The permutation change caused by aligning $\mathbf{W}_{b-1}^{(i)}$ to $\mathbf{W}_{b-1}^{(i+1)}$ disrupts the original neuron correspondence between $\mathbf{W}_{b-1}^{(i)}$ and $\mathbf{W}_b^{(i)}$. Such permutation change information is stored in $\mathbf{T}_{\text{in}}$ for $\mathbf{W}_b^{(i)}$. To restore this, $\mathbf{W}_b^{(i)}$ needs to align with $\mathbf{W}_{b-1}^{(i)}$ using $\mathbf{T}_{\text{in}}$. $\mathbf{T}_{\text{in}}$ is defined by $\mathcal{TMF}$ for each module, shown in Table 1. After retrieving $\mathbf{T}_{\text{in}}$ (cf. line 5), the alignment within the layer is performed via $\mathbf{W}_b^{(i)} \leftarrow \mathbf{W}_b^{(i)} \cdot \mathbf{T}_{\text{in}}$ (cf. line 6).

**Step-3: Alignment across layer** We then solve $\text{OT}(\mu, \nu, \mathbf{C})$ to compute the transport matrix. Imfeld et al. (2024) found that the Sinkhorn-Knopp algorithm Knight (2008) is optimal for solving $\text{OT}(\mu, \nu, \mathbf{C})$ in Transformer fusion. We employ this approach to obtain $\mathbf{T}_{\text{out}}$ for $\mathbf{W}_b^{(i)}$ (cf. line 7). $\mathbf{W}_b^{(i)}$ is then aligned with $\mathbf{W}_b^{(i+1)}$ using the computed $\mathbf{T}_{\text{out}}$ via $W_b^{(i)} \leftarrow \mathbf{T}_{\text{out}}^{\top} \cdot W_b^{(i)}$ (cf. line 8).

**Step-4: Compute new-layer weights.** $\mathbf{W'}_b^{(i)}$ is the average of aligned $\mathbf{W}_b^{(i)}$ and $\mathbf{W}_b^{(i+1)}$ (cf. line 9).

**Step-5: Zero-Initialization.** We set $\mathbf{T}_{in} = \mathbf{I}$ for $\mathbf{W}_O$ and $\mathbf{W}_{down}$ in TMF due to architectural advancements (Section 4.2), which may cause misalignment problem. Inspired by zero-initialization (Wu et al., 2024), we set $\mathbf{W}_O = 0$ and $\mathbf{W}_{down} = 0$ (cf. line 11), which naturally resolves this issue while ensuring function preservation, a property crucial for retaining model performance (Wang et al., 2024; Wu et al., 2024).

## 5 Experimental Setup

### 5.1 Base Model

Following prior work (Wu et al., 2024; Kim et al., 2024; Yang et al., 2025), we use the off-the-shelf 32-layer Llama-3.1-8B (Grattafiori et al., 2024) as our *base* model. We further conducted a smaller-scale experiment using the off-the-shelf 16-layer Llama-3.2-1B.

### 5.2 Baselines

We experiment with state-of-the-art depth up-scaling methods, as shown in Figure 1. Following Yang et al. (2025), we insert a number of new layers equal to 50% of the base layers. The *expanded* model sizes are fixed at 11.5B parameters with 48 layers (adding 16 layers) and 1.72B with 24 layers (adding 8 layers) for all depth up-scaling methods.

**Base.** We continue pre-training the *base* model without expansion. All layers are trained.

**SOLAR.** This method copies the bottom and top $m$ layers from $\mathcal{M}$ to form $\mathcal{M}'$. We choose $m = 24$ and $m = 12$ for 11.5B and 1.72B *expanded* models. All layers are trained in line with Kim et al. (2024).

**LLaMA PRO.** It divides $\mathcal{M}$ into $g$ groups of $m$ layers. $p$ new layers are created by copying the top-$p$ base layers and inserted on top of each group. These new layers are initialized with $W_O = W_{down} = 0$. We use $g = 16$ for the 11.5B *expanded* models and $g = 8$ for the 1.72B *expanded* models; $m = 2$ and $p = 1$ are used throughout. Only $f'$ are trained following Wu et al. (2024).

**LESA.** This approach uses an auxiliary network to initialize $f'_i$ given $f_i$ and $f_{i+1}$. LESA inserts $f'_i$ in the top half of $\mathcal{M}$. We insert new layers between $f_{16}$ and $f_{32}$ for the 11.5B *expanded* models, and between $f_8$ to $f_{16}$ for the 1.72B *expanded* models. Only $f'$ are trained as in Yang et al. (2025).

### 5.3 Training Data

For Continual Pre-Training (CPT), we opt using data of same size as in Yang et al. (2025), published after the base model's knowledge cut-off. We sample 1.5B tokens from the CC-MAIN-2024-51 subset of FineWeb-Edu (Penedo et al., 2024). For supervised fine-tuning (SFT), we choose Alpaca GPT4 (Peng et al., 2023) and update the whole model following Yang et al. (2025).

### 5.4 Evaluation

Following previous studies (Wu et al., 2024; Yang et al., 2025), we conduct experiments focusing on **general** knowledge-related tasks. We further conduct extensive experiments on specialized domains: **biomedical** (Lee et al., 2019; Gu et al., 2021; Luo et al., 2022; Singhal et al., 2022; 2023) and **legal** (Chalkidis et al., 2019; Zheng et al., 2021; Henderson et al., 2022; T.y.s.s et al., 2024; Niklaus et al., 2024) as they are widely explored within LLMs.

**General.** We include ARC-Easy (Clark et al., 2018), LogiQA (Liu et al., 2020), Winogrande (Sakaguchi et al., 2021) for **Reasoning**; CSQA (Talmor et al., 2019), BoolQ (Clark et al., 2019), PIQA (Bisk et al., 2020) for **Commonsense and Knowledge**; MMLU (Hendrycks et al., 2021) for **Examination**; and WikiText (Merity et al., 2017) for **Language Modeling**.

**Biomedical.** Following previous work (Williams et al., 2025), we include the MultiMedQA benchmark (Singhal et al., 2022), specifically the PubMedQA (Jin et al., 2019), MedQA (Jin et al., 2021), MedM-

| | Methods | Perplexity↓ | Zero-shot Performance↑ | | | | | | | |
|---|---|---|---|---|---|---|---|---|---|---|
| | | Wiki-PPL | ARC | LogiQA | Wino | CSQA | BoolQ | PIQA | MMLU | Average |
| **CPT** | Base-8B | 8.35 | 79.97 | 26.88 | 72.06 | 65.19 | 81.83 | 78.84 | 58.61 | 66.20 |
| | SOLAR-11.5B | 9.90 | 79.88 | 26.88 | 71.59 | 57.41 | 80.70 | 78.56 | 54.37 | 64.20 |
| | LLaMA PRO-11.5B | 7.81 | 81.61 | **29.49** | 73.72 | 70.93 | 81.65 | 79.98 | 62.56 | 68.56 |
| | LESA-11.5B | 7.73 | **82.07** | 27.96 | 74.11 | **72.40** | 81.93 | 80.30 | 62.63 | 68.77 |
| | OpT-DeUS-11.5B (Ours) | **7.73** | **82.07** | 27.34 | **74.74** | 71.91 | **82.26** | **80.79** | **62.96** | **68.87** |
| **SFT** | Base-8B | 8.32 | 81.10 | 24.58 | 72.14 | 68.30 | 82.14 | 79.71 | 59.17 | 66.73 |
| | SOLAR-11.5B | 9.68 | 80.68 | 25.19 | 71.19 | 61.18 | 81.19 | 79.16 | 55.03 | 64.80 |
| | LLaMA PRO-11.5B | 7.81 | 83.33 | **27.19** | 74.11 | 72.07 | 82.26 | 80.79 | 62.32 | 68.87 |
| | LESA-11.5B | **7.72** | **83.84** | 26.57 | 75.53 | **73.05** | 83.00 | 80.69 | 63.57 | 69.47 |
| | OpT-DeUS-11.5B (Ours) | 7.73 | 83.80 | 26.73 | **76.09** | **73.05** | **83.36** | **80.85** | **63.84** | **69.67** |
| **CPT** | Base-1B | 13.68 | 68.64 | 21.35 | 58.48 | 24.57 | 62.32 | 74.97 | 28.85 | 48.46 |
| | SOLAR-1.72B | 13.87 | **68.90** | 21.20 | 59.67 | 21.21 | 61.07 | 74.76 | 28.58 | 47.91 |
| | LLaMA PRO-1.72B | 12.43 | 67.26 | 21.04 | **61.96** | 34.48 | 62.91 | **75.52** | 31.85 | 50.72 |
| | LESA-1.72B | 12.28 | 66.71 | 21.20 | 59.75 | 41.03 | **63.64** | 74.76 | **33.47** | 51.51 |
| | OpT-DeUS-1.72B (Ours) | **12.19** | 67.00 | **22.58** | 60.77 | **43.00** | 62.72 | 75.03 | 33.02 | **52.02** |
| **SFT** | Base-1B | 13.57 | 69.87 | **22.43** | 59.43 | 26.29 | 62.81 | 75.57 | 29.91 | 49.47 |
| | SOLAR-1.72B | 13.68 | **70.41** | 22.27 | 59.27 | 24.90 | 60.83 | 75.84 | 29.40 | 48.99 |
| | LLaMA PRO-1.72B | **12.36** | 68.14 | 21.35 | 60.30 | 38.08 | 64.07 | **76.12** | 30.73 | 51.26 |
| | LESA-1.72B | 12.54 | 67.76 | 20.89 | 59.98 | 43.73 | 64.86 | 75.84 | **34.47** | 52.51 |
| | OpT-DeUS-1.72B (Ours) | 12.46 | 68.31 | 21.51 | **60.46** | **44.47** | **65.84** | 75.84 | 33.16 | **52.80** |

Table 2: CPT on 1.5B tokens and SFT (after CPT) performance of 11.5B and 1.72B *expanded* models.

CQA (Pal et al., 2022) tasks, and relevant subsets from MMLU (Hendrycks et al., 2021) (anatomy, clinical knowledge, college medicine, medical genetics, professional medicine, college biology).

**Legal.** We follow Williams et al. (2025) in using CaseHOLD (Zheng et al., 2021) and ECtHR (Task A) (Chalkidis et al., 2019) datasets from the LexGLUE benchmark (Chalkidis et al., 2022) and Legal-MMLU, covering jurisprudence, professional law, and international law specialties (Hendrycks et al., 2021).

### 5.5 Hyper-parameter Details

We set the regularization parameter of Sinkhorn-Knopp algorithm to 0.06, as in Imfeld et al. (2024). We set the global batch size and sequence length to 64 and 2048. For CPT, we use a maximum learning rate of 1e-4 for 1.72B *expanded* models and 5e-5 for 11.5B *expanded* models. For SFT, the maximum learning rate is set to 1e-5 and 5e-6, respectively.

### 5.6 Implementation Details

We employ Flash-Attention 2 (Dao, 2024) and mixed-precision `bf16` for accelerated training. We use Language Model Evaluation Harness (Gao et al., 2024) for evaluation. 11.5B *expanded* models are trained on four NVIDIA GH200 (96GB) GPUs while 1.72B *expanded* models are trained on a single NVIDIA A100 (80GB). We create all *expanded* models using AMD EPYC 7413 CPU and a single NVIDIA A100 (80GB).

## 6 Experimental Results

### 6.1 General Performance

Table 2 (Top) presents the CPT and SFT results of our 11.5B *expanded* models. For CPT, we observe that OpT-DeUS achieves top performance on six out of eight benchmarks, specifically Wiki-PPL (7.73), ARC (82.07) ,Winogrande (74.74), BoolQ (82.26), PIQA (80.79), MMLU (62.96). Furthermore, OpT-DeUS ranks second on CSQA. This strong performance across various downstream tasks, resulting in the highest

average score (68.87), highlights the effectiveness of our approach. We further note that OpT-DeUS's strong performance continues in SFT. It achieves top performance on Winogrande, CSQA, BoolQ, PIQA, MMLU and second performance on Wiki-PPL, ARC and LogiQA, yielding the highest average score (69.67).

To further analyze performance during training, we save five checkpoints while training the 11.5B *expanded* models (20%, 40%, 60%, 80% and 100% of training steps), shown in Appendix A. We observe that OpT-DeUS consistently achieves top performance on at least five out of eight benchmarks across all checkpoints regardless the size of the CPT data.

## 6.2 Domain Specific Performance

| Methods | Biomedical | | | | Legal | | |
|---|---|---|---|---|---|---|---|
| | MedMCQA | MedQA | Bio-MMLU | PubMedQA | Legal-MMLU | CaseHOLD | ECtHR |
| Base | 48.77 | 53.26 | 68.45 | 76.20 | 64.49 | 47.42 | 56.19 |
| SOLAR | 45.09 | 47.29 | 57.54 | 76.20 | 62.60 | 41.17 | 39.11 |
| LLaMA PRO | 54.72 | 59.47 | 70.88 | **77.80** | **68.41** | 48.50 | 60.27 |
| LESA | 56.51 | 59.86 | **71.73** | 76.00 | 67.02 | 51.08 | 60.89 |
| OpT-DeUS | **56.63** | **60.25** | 71.03 | 76.20 | 67.93 | **51.83** | **61.26** |

Table 3: Domain-specific CPT Performance of 11.5B *expanded* models.

Table 3 presents the CPT results of 11.5B *expanded models* on biomedical and legal domains. We observe OpT-DeuS achieves the best overall performance. In biomedical tasks, OpT-DeuS wins two out of four tasks (i.e. MedMCQA and MedQA), while offering the second best performance on the remaining two. Strong performance is also observed in the legal domain, where OpT-DeUS wins two out three legal tasks and achieves second-best on the remaining one.

## 6.3 Performance at Smaller Scales

Table 2 (Bottom) presents the CPT and SFT results of 1.72B *expanded* models. For CPT, OpT-DeUS achieves the best overall performance (52.02) and ranks first on Wiki-PPL (12.19), LogiQA (22.58), and CSQA (43.00), while ranking second on Winogrande, PIQA, and MMLU. Compared to LESA, the second-best method, OpT-DeUS obtains the highest average score (52.02 vs. 51.51) and achieves top-2 performance on most downstream tasks (6 vs. 4). For SFT, strong performance can still be observed with the highest average score. OpT-DeUS wins on Winogrande, CSQA, and BoolQ, while being second on Wiki-PPL, PIQA and MMLU. Similar to the results of the 11.5B *expanded* models, OpT-DeUS is the best-performing method using a smaller *base* model. This consistency demonstrate OpT-DeUS's robustness to model sizes.

Interestingly, we find SOLAR obtains poor performance on both sizes. For example, it performs worse than the *base* model (Avg: 64.20 vs 66.20; 47.91 vs 48.46). We hypothesize that SOLAR's poor performance is caused by catastrophic forgetting. Fully updating the *expanded* model substantially degrades the pre-trained parametric knowledge.

## 6.4 Up-scaling Stability at Larger Scales

We follow previous work (Yano et al., 2025; Yang et al., 2025) by reporting perplexity without any model training to evaluate up-scaling stability on larger models. Appendix B presents the perplexity at different model scales. We observe that both LLaMA-Pro and OpT-DeUS match the base model's perplexity regardless of model parameters due to zero-initialization, demonstrating maximum expansion stability compared to other baselines. Surprisingly, we find that LESA's perplexity sharply increases when applied to Llama-3.2-1B (871.50). We hypothesize this is because smaller models have fewer layers. This leads to less training data for the auxiliary network, consequently causing it to underfit.

# 7 Experimental Analysis

## 7.1 Interpolation Positions

We conduct an ablation study on OpT-DeUS to determine the best interpolation approach. We evaluate the following strategies: inserting in the bottom half (Btm), in the middle portion (Mid), in the top half (Top), and at the top and bottom quarters (T&B). The layer index ranges are defined as follows:

$$\mathcal{M}'(x;\theta')\circ = \begin{cases} f'_i \circ f_i, & i \leq \frac{n}{2} & \text{if Btm} \\ f'_i \circ f_i, & \frac{n}{4} < i \leq \frac{3n}{4} & \text{if Mid} \\ f'_i \circ f_i, & \frac{n}{2} \leq i < n & \text{if Top} \\ f'_i \circ f_i, & i \leq \frac{n}{4} \text{ or } \frac{3n}{4} \leq i < n & \text{if T&B} \end{cases}$$

Table 4 illustrates the performance of different interpolation strategies. We observe that OpT-DeUS-Top is the best performing strategy, overall. OpT-DeUS-Top yields the highest average performance (68.87), winning in six out of eight benchmarks (i.e. ARC, Winogrande, CSQA, BoolQ, PIQA, MMLU). The performance difference between interpolation strategies is consistent with previous work, where inserting new layers into the top part offers additional performance gains (Yang et al., 2025). This phenomenon further supports previous findings showing that bottom layers in Transformers are more critical (Jawahar et al., 2019), while top layers are less sensitive to modification (Men et al., 2025).

| | Perplexity↓ | Zero-shot Performance↑ | | | | | | | |
|---|---|---|---|---|---|---|---|---|---|
| Methods | Wiki-PPL | ARC | LogiQA | Wino | CSQA | BoolQ | PIQA | MMLU | Average |
| OpT-DeUS-Btm | 7.83 | 81.69 | _28.26_ | 74.35 | 70.02 | 81.74 | 79.92 | 62.28 | 68.32 |
| OpT-DeUS-Mid | **7.70** | **82.07** | 27.65 | 74.35 | _70.11_ | 81.07 | _80.25_ | _62.56_ | 68.29 |
| OpT-DeUS-Top | _7.73_ | **82.07** | 27.34 | **74.74** | **71.91** | **82.26** | **80.79** | **62.96** | **68.87** |
| OpT-DeUS-T&B | 7.87 | 81.40 | **28.57** | _74.51_ | 70.02 | _82.11_ | 79.87 | 62.46 | _68.42_ |

Table 4: Performance of 11.5B OpT-DeUS trained on 1.5B tokens using different interpolation strategies.

## 7.2 Training Efficiency

Previous work ignores the impact of interpolation strategy regarding training efficiency (Wu et al., 2024; Yang et al., 2025). Table 5 shows that progressive depth up-scaling methods considerably outperform SOLAR (22:54:11) in training efficiency. We observe a strong correlation between interpolation positions and efficiency: top-half insertions, exemplified by OpT-DeUS-Top (12:52:04) and LESA (12:54:07), are notably faster. Conversely, strategies inserting layers in the bottom half, such as OpT-DeUS-Btm (14:56:00) and LLaMA PRO (14:58:34), require

Table 5: Training time for 11.5B models.

| Methods | Trainable | Total | Training Time |
|---|---|---|---|
| SOLAR | 11B | 11.5B | 22:54:11 (+78.0%) |
| LLaMA PRO | 4B | 11.5B | 14:58:34 (+16.4%) |
| LESA | 4B | 11.5B | 12:54:07 (+0.3%) |
| OpT-DeUS-Btm | 4B | 11.5B | 14:56:00 (+16.1%) |
| OpT-DeUS-Mid | 4B | 11.5B | 13:53:14 (+7.9%) |
| OpT-DeUS-Top | 4B | 11.5B | 12:52:04 |
| OpT-DeUS-T&B | 4B | 11.5B | 14:45:38 (+14.7%) |

longer training time. This pattern persists regardless of the weight initialization method. The observed efficiency differences are primarily due to increased back-propagation costs when updating new layers inserted at lower model positions.

Both LESA and OpT-DeUS require additional computation. LESA necessitates extracting latent patterns using Singular Value Decomposition (SVD) to train an auxiliary fixed-size neural network, while OpT-DeUS requires solving the OT problem module-wise. Table 6 presents the time required for LESA and OpT-DeUS to create and train the *expanded* model. Note that the training time difference between the 1.72B *expanded* and 11.5B *expanded* models is due

Table 6: Instantiation time for LESA and OpT-DeUS.

| Expanded Model | Training Time | Creating Time |
|---|---|---|
| LESA 1.72B | 31:08:17 | 00:26:15 |
| OpT-DeUS 1.72B | 30:58:56 | 00:02:34 |
| LESA 11.5B | 12:54:07 | 04:52:13 |
| OpT-DeUS 11.5B | 12:52:04 | 00:37:16 |

to the different hardware used (i.e. one A100 vs. four GH200) for training. We observe that LESA requires more time compared to OpT-DeUS (00:26:15 vs. 00:02:34). This time scales massively with larger models (04:52:13 vs. 00:37:16). We hypothesize that this increased time for LESA is mainly caused by the extra computation required for SVD when scaling up base models. Combining training and creation times across different scales of base models, our OpT-DeUS achieves the best time efficiency among the baselines.

## 7.3 Ablation Studies

| | Methods | Perplexity↓ | | Zero-shot Performance↑ | | | | | | |
|---|---|---|---|---|---|---|---|---|---|---|
| | | Wiki-PPL | ARC | LogiQA | Wino | CSQA | BoolQ | PIQA | MMLU | Average |
| **TMF** | Attention Output $\mathbf{T}_{\text{in}} = \mathbf{T}_Q$ | **12.05** | 67.13 | **23.35** | 60.30 | 42.67 | 62.87 | 75.41 | 32.10 | 51.98 |
| | MLP Gate and Up $\mathbf{T}_{\text{in}} = \frac{1}{2}(\mathbf{T}_O + \mathbf{I})$ | 12.09 | 66.84 | 22.89 | 60.38 | 42.59 | 62.23 | **75.46** | 32.24 | 51.80 |
| **OT Impact** | Average | 12.62 | 67.72 | 22.12 | 59.19 | 39.23 | 62.51 | 74.65 | 30.72 | 50.88 |
| | Copy | 12.62 | **68.01** | 22.73 | 59.67 | 34.81 | 62.97 | 74.43 | 28.25 | 50.12 |
| | Random | 16.40 | 66.62 | 22.27 | 59.67 | 40.79 | 62.72 | 74.97 | 32.19 | 51.32 |
| | Element-wise Shuffle after OT | 12.05 | 67.34 | 21.51 | 59.35 | 43.00 | 64.16 | 74.59 | 33.23 | 51.88 |
| **Zero-Init** | Average + Zero-Init | 12.62 | 67.72 | 22.12 | 59.19 | 39.23 | 62.51 | 74.65 | 30.72 | 50.88 |
| | Copy + Zero-Init | 12.23 | 67.13 | 22.43 | 59.51 | 40.87 | **64.28** | 75.03 | 32.42 | 51.67 |
| | Random + Zero-Init | 12.20 | 66.92 | 21.51 | 59.43 | 42.92 | 62.17 | 74.81 | **33.32** | 51.58 |
| | OpT-DeUS (Ours) | 12.19 | 67.00 | 22.58 | **60.77** | **43.00** | 62.72 | 75.03 | 33.02 | **52.02** |

Table 7: Ablation study on 1.72B *expanded* models. Notably, Copy+Zero-Init corresponds to initialize new layers using LLaMA PRO but interpolate at different positions.

**Ablation on TMF design.** In Section 4.2, we modify the vanilla TMF because architectural advancements (i.e., Attention Output and MLP Down $\mathbf{T}_{\text{in}} = \mathbf{I}$) and non-consecutive alignment (i.e, MLP Gate and Up $\mathbf{T}_{\text{in}} = \mathbf{T}_O$). We further compare this choices with other applicable variants for architectural advancements (i.e., Output $\mathbf{T}_{\text{in}} = \mathbf{T}_Q$) and non-consecutive alignment (i.e, Gate and Up $\mathbf{T}_{\text{in}} = \frac{1}{2}(\mathbf{T}_O + \mathbf{I})$).

Table 7 (Top) present the performance when using different TMF choices, we observe that our OpT-DeUS yield the best overall performance (Avg: 52.02). This indicates that our TMF modification on both architectural advancements and non-consecutive alignment is valid and provide extra improvements.

**Ablation on OT alignment.** OpT-DeUS introduce OT-alignment for aligning mismatched neurons. To validate that OT alignment do mitigate neuron permutation mismatch (Section 4.1), we compare it with standard copy,average and random initialization. We further compare against a variant that randomly shuffles weights element-wise after OpT-DeUS initialization (i.e., Element-wise Shuffle after OT), which disrupts the permutation structure and neuron-level correspondences established by OT alignment.

Table 7 (Middle) present the corresponding performance. We observe that OpT-DeUS achieves the top performance. Interestingly, Element-wise Shuffle after OT yields better performance than non-OT variants(i.e., Average, Copy and Random), as OT alignment captures neuron-level functional similarity, leading to more informative construction of new neurons. Comparing OpT-DeUS with average, and Element-wise Shuffle after OT that explicitly introduce neuron permutation mismatch, the better performance of OpT-DeUS validates that OT-alignment provide meaningful initialization that mitigate neuron permutation mismatch.

**Ablation on Zero-Initialization.** OpT-DeUS adopts zero-initialization to mitigate potential alignment issues (Section 4.3). We construct non-OT variants with zero-initialization, as shown in Table 7 (Bottom). OpT-DeUS consistently outperforms the corresponding baselines. Moreover, comparing Average, Copy, and Random initialization with their zero-initialized variants shows that zero-initialization improves performance. Together with the observed up-scaling stability (Section 6.4), these results further validate the effectiveness of zero-initialization in OpT-DeUS.

### 7.4 Analysis of Neuron Functionality

To investigate the neuron functionality mapping between base layers and new layers, we analyse the transport matrix for each module between each new layer and its two adjacent base layers. Following Algorithm 1 (Section 4.3), we first instantiate the OT problem and using Sinkhorn–Knopp algorithm (same hyper-parameters) to solve it. We then compute the Shannon entropy (Shannon, 1948) for each transport matrix.

We report the layer-averaged entropy of transport matrix for each module (detailed in Table 1) within the Transformer layer in Table 8. Transport matrix indicates the neuron-level functionality correspondence between layers, while the entropy evaluates how widely the mapping is distributed. A higher entropy of the transport matrix indicates a more diffuse and smoother mapping (Cuturi, 2013). Notably, SOLAR and LESA are excluded because they are inapplicable: SOLAR adds a continuum of new layers, and LESA does not directly use neurons from base layers.

| | Attention | | | | MLP | | | |
|---|---|---|---|---|---|---|---|---|
| Methods | Query | Key | Value | Output | Gate | Up | Down | Average |
| LLaMA PRO | 12.23 (12.25) | 9.99 (10.00) | 10.29 (10.39) | 16.64 (16.63) | 14.17 (14.18) | 14.20 (14.27) | 16.63 (16.63) | 13.45 (13.48) |
| Average | 11.69 (11.76) | 7.32 (7.33) | 10.90 (11.03) | 15.60 (15.73) | 17.06 (17.58) | 18.71 (18.88) | 8.56 (8.83) | 12.84 (13.02) |
| OpT-DeUS | 13.25 (13.28) | 10.32 (10.32) | 11.57 (11.66) | 16.64 (16.64) | 16.78 (17.29) | 18.36 (18.67) | 16.64 (16.63) | 14.79 (14.93) |

Table 8: Layer-averaged Shannon Entropy of the transport matrix between base layers and new layers for 11.5B *expanded* models. Values outside parentheses indicate entropy *before* training, while values inside parentheses represent entropy *after* training. Note that normalization components are excluded, as they only apply element-wise scaling without cross-neuron information mixing.

The marginal change in entropy before and after training is expected, as entropy reflects the neuron-level functionality mapping established at new-layer instantiation rather than being learned during training. We observe a clear correlation between high transport matrix entropy and strong performance. OpT-DeUS achieves the highest entropy (14.79) and the strongest performance (68.87), followed by LLaMA PRO (13.45/68.56) and Average (12.84/68.39). From the perspective of information theory, high entropy indicates that the information from each old neuron is distributed across multiple neurons in next layer. This suggests that $f_i'$ layer forms representations through a richer mixture of input features, integrating information to construct a more diverse representational space (Tax et al., 2017; Yu et al., 2021). Such distributed representations improve expressive capacity, aligning with the observed performance gains.

Interestingly, we found the entropy of the Key Projection (7.32 vs 9.99/10.32) and Down Projection (8.56 vs 16.63/16.64) is low when using direct average for initialization. This finding suggests averaging the Key Projection and Down Projection are more sensitive thus leading to greater performance degradation. This low entropy suggests that the functionality from each old neuron is concentrated on fewer neurons in the new lyaer. As a result, this small subset of neurons implements a large fraction of the module's functionality, thereby dominating the module output. This is further consistent recent work showing the low-rank bottleneck in Query and Key Projection (Bhojanapalli et al., 2020) and the parameter redundancy in Down Projection (Pires et al., 2023; Wei et al., 2024).

## 8 Conclusion

We introduced OpT-DeUS, a progressive depth up-scaling approach using OT. Our approach conducts neuron alignment within and across layers to mitigate the neuron permutation mismatch. Empirical results demonstrate that OpT-DeUS offers better downstream performance with improved training efficiency than other depth up-scaling approaches. Our extensive experiments verify the effectiveness of OpT-DeUS on both continual pre-training and supervised fine-tuning across different model scales and diverse downstream tasks. Our analysis of interpolation positions reveals their impact on training efficiency, demonstrating that inserting new layers closer to the top leads to higher training efficiency due to shorter back-propagation paths through the trainable new layers. Our entropy analysis further reveals the correlation between strong performance and high transport matrix entropy when initializing new layers.

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

## A    Performance across checkpoints

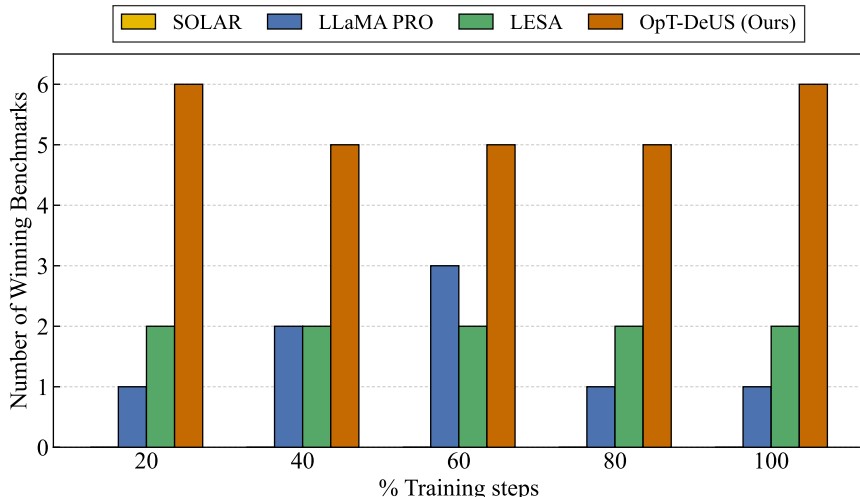

Figure 3: Number of benchmarks that achieve top performance during the training process of 11.5B *expanded* models. Sums may exceed 8 due to ties.

## B    Scaling Stability

| Model | Base | SOLAR | LLaMA PRO | LESA | OpT-DeUS |
|---|---|---|---|---|---|
| Llama-3.2-1B | 11.57 | 16.64 | 11.57 | 871.50 | 11.57 |
| Llama-3.1-8B | 7.33 | 9.01 | 7.33 | 9.35 | 7.33 |
| Mistral-24B | 4.43* | 6.51* | 4.43 | 5.17* | 4.43 |
| Qwen-2.5-32B | 3.78* | INF* | 3.78 | 5.67* | 3.78 |
| Llama-3-70B | 1.98* | 4.21* | 1.98 | 2.62* | 1.98 |

Table 9: PPL after 1.5x layer expansion initialization for different *base* models, along with PPL of base models. * denotes results from Yang et al. (2025), Results for LLaMA PRO and OPT-DeUS in bottom-half are obtained via reasonable extrapolation from top-half.

