# OpenReview forum: "Progressive Depth Up-scaling via Optimal Transport"
_TMLR — Rejected by TMLR_

### Review · Reviewer_C481 · 2025-12-21

**Summary Of Contributions:**

The paper proposes a novel method for depth up-scaling, i.e., the practice to extend the depth of a pre-trained model without retraining the whole model from scratch. The paper addresses the problem of how to initialize the inserted layers and where to insert these layers. In particular, inspired by previous work that initializes new layers as the interpolation of the parameter between the adjacent layers, they address the problem of permutation-invariance that is not accounted for in simple interpolation (averaging). The authors apply a method based on Optimal Transport, previously used for transformer model merging, to align adjacent blocks before averaging them. Experimentally, on a fair range of evaluations and with remarkable transformer sizes, they show that their method outperforms simple averaging, as well as previous work’s methods. Finally, they also show that inserting new layers closer to the output, initialized with their method, achieves lower training time with better performance.

### Strengths:

The idea of accounting for neuron permutations before averaging, and using that as a starting point before finetuning, makes sense, and could also serve as an interesting test bed for studying internal representation and signal propagation in transformers. The paper reads well, and the graphics are visually appealing. Overall, I find the ideas promising.

### Weaknesses:

I find that some crucial choices in the alignment strategies presented in Fig. 3 are not supported by explanations or intuitions, and therefore appear as arbitrary. Furthermore, these choices break the purpose of the transport matrix flow propagation proposed in (Imfeld et al. 2024), namely, to preserve the alignment consistency through the model. One can argue that, in general, strong empirical performance can serve as a sufficient support for arbitrary methodological choices; I personally agree with this view. However, I find that also the experimental setup (as currently presented) is highly problematic: the choice of hyperparameters, and in particular of the learning rate, seems absolutely arbitrary. Most importantly, the same hyperparameters are shared across all methods, possibly confounding the results and heavily favoring the proposed method. This, in particular, is not clear if the LR used is tuned specifically for the proposed method. Good experimental practice requires at least that each method can independently pick the best hyperparameters on a (not necessarily large) grid of options. To obviate computational limitations, choosing smaller models would have allowed a thorough and fair empirical evaluation of all models.

**Audience:**

Yes

**Audience Explanation:**

The idea of the paper is rather incremental, but sound, and I am sure it can be of interest to the community. Also, analyzing adjacent blocks of transformers through the lens of OT-based alignment is a clean test-bed for studying internal representation and signal propagation.

**Broader Impact Concerns:**

None.

**Claims And Evidence:**

No

**Claims Explanation:**

As mentioned in the summary, I believe that the current version of the paper presents two problems that heavily impact the overall support of the claims, despite being promising. I will elaborate further on these below, along with some other comments.

Of course, should the author be able to provide strong intuitions/explanations of the methodological choices, together with a thorough experimental verification that crucially shows that the nature of the gains is indeed the alignment, I will be happy to reconsider my review.

---

## Alignment choices

The idea of the transport map flow diagrams of (Imfeld et al. 2024) is to allow preserving the consistency of the aligned model, i.e., to propagate the previous alignments to later layers before aligning these later layers. The problem I have is that the authors propose propagation strategies that I find arbitrary and not supported by sufficient explanation or intuition.

First of all, while I understand the reasoning of zero-init of certain layers (to preserve network invariance right after inserting the new layers), this choice defies the spirit of the transport matrix flow and the alignment itself, as it “breaks” the overall consistency of the layers. Very importantly (also for the experimental verification), this choice is **not** applied to the Avg strategy, representing an arbitrary choice which potentially is a strong confounder for the experimental results.

Secondly, and most importantly, I don’t agree with the choice of setting $T_{in}=I$ for the output layer of the attention module. This implies that the output layer alignment $T_O$ is ignoring the incoming permutation from the attention layers, and therefore, the outgoing map $T_O$ has hardly anything to do with the actual permutation that needs to be accounted for. This map, furthermore, is propagated to the RMSNorm (rightly combined with the residual stream), meaning that this faulty $T_O$ propagates the error further. Finally, I don’t understand why Gate and Up get only $T_O$ as input, and not the combination of $T_O$ and residual streams.

These choices are, in my opinion, not sufficiently justified, neither intuitively nor empirically (e.g., with ablations).

---

## Experimental Setup

As mentioned above, I believe that independent tuning of the hyperparameters on a small (e.g., 3 LR) grid is absolutely necessary to have convincing results (especially as the intuitive explanations are insufficient). Right now, the choice of LRs seems arbitrary, or worse: possibly tuned on the OPT strategy, and then transferred to the other methods. This is not ruled out by the text, and is therefore highly problematic.

Secondly, as mentioned, the fact that the Avg strategy does not have the zero-init also impedes a fair assessment of whether the alignment itself is the reason for the gains in performance. Finally, I would have appreciated a baseline with random init (and zero init on the same layers) as a comparison.

For these reasons, I believe the claims attributing the performance benefits to the alignment are not sufficiently supported by empirical evidence.

---

## Other.

### Layers vs. Blocks

I find it highly confusing that layers are the “transformer layers” and blocks are the “inner matrices”. This is in contrast to the usual nomenclature in the transformer literature, where blocks are the “outer structure”. I would suggest flipping the names altogether.

### Figure 1

Figure 1 is visually appealing, though not clear what the OpT-DeUS actually does, and it is actually rather confusing; in particular, it is not clear what colors stand for.
There is a typo “alinged” (bottom right).

### Notation

Matrices should use bold symbols. Also, Figures should use the same notation as in 3.1 for clarity. Also, in 3.1, h is not introduced.

### Figure 2

Better state that the right part shows two different strategies, and which is which.

### Algorithm

An algorithm should be self-contained, i.e., it should introduce all symbols and variables. This is not the case, and it is therefore impossible to understand without reading through the text.

### Sinkhorn

I would have liked some more information about the Sinkhorn algorithm, i.e., its intuitive idea that it finds a “soft” alignment, rather than a hard one.

### Base model

It is not clear if the model used in the experiments is pretrained off-the-shelf or if the authors trained it from scratch before the experiments.

### 5.4

Typo “specialzed”.

### Sinkhorn regularizer

I would have highly appreciated an ablation on the role of the Sinkhorn regularizer in the final result.

### Performance at Larger Scales

I think Table 3 is misleading and does not give any insights: obviously, OpT-DeUS and LLaMA Pro preserve performance at initialization, so I do not see how this is informative for the final performance. Finally, I find extrapolation hard to justify.

### Training Efficiency

Typo “ingones”.

I find it uncommon to report training times, as they are so dependent on actual hardware and implementation. This is confirmed by the fact that some models exhibit much higher training time when run on other hardware. Measuring FLOPs would be a better metric, or at the very least, use the same hardware, or find hardware-independent normalization. Overall, I find this section quite trivial, though; I would probably move it to the appendix.

### Neuron Functionality

Analyzing the nature of the alignment is very interesting. However, there are insufficient details on how the experiments are executed. How is the entropy calculated? Does it depend on the Sinkhorn regularizer used for the alignment? Is it somehow dependent on the dimensionality of the output? How do you account for that? I find it quite odd that the metric varies so little before and after training.

I am not sure I follow why low entropy suggests a smaller effective rank.

**Requested Changes:**

The full list of changes or clarifications needed is above.

To support the acceptance of the paper, concretely, I need (i) further explanations on the doubts regarding the strategy of alignment, and, most importantly, (ii) empirical evaluation with fair hyperparameter tuning, as well as (iii) better baselines that allow to convincingly identify the OT-alignment as the reason for the performance improvements (e.g., avg with zero-init, random init with zero-init, etc.). I suggest that the authors focus on these crucial changes first.

Secondary changes: the final section on neuron functionality needs better explanation and more details; typos; refinement of figures to better convey the strategies; and algorithmic notation.

---

> ### Author Response · Authors · 2026-01-27
> **Thanks for your constructive review.**
>
> We appreciate your constructive feedback. Below, we provide our point-to-point responses.
>
> ***Weaknesses***
>
> > I find that some crucial choices in the alignment strategies presented in Fig. 3 are not supported by explanations or intuitions, and therefore appear as arbitrary. Furthermore, these choices break the purpose of the transport matrix flow propagation proposed in (Imfeld et al. 2024), namely, to preserve the alignment consistency through the model. One can argue that, in general, strong empirical performance can serve as a sufficient support for arbitrary methodological choices; I personally agree with this view.
>
> In the revised version of the paper, we additionally conduct extensive ablation studies in Section 7.3, covering all variants suggested from you in **Alignment Choices** and **Requested Changes** section. OpT-DeUS consistently achieves the best performance across all ablation variants, validating our design choices.
>
> > However, I find that also the experimental setup (as currently presented) is highly problematic: the choice of hyperparameters, and in particular of the learning rate, seems absolutely arbitrary. Most importantly, the same hyperparameters are shared across all methods, possibly confounding the results and heavily favoring the proposed method. This, in particular, is not clear if the LR used is tuned specifically for the proposed method. Good experimental practice requires at least that each method can independently pick the best hyperparameters on a (not necessarily large) grid of options. To obviate computational limitations, choosing smaller models would have allowed a thorough and fair empirical evaluation of all models.
>
> We use the training hyperparameters aligning with the configuration in LESA [1]. Similarly, for all up-scaling baselines, we use the same hyperparameters from the respective original work to up-scale models, varying only the number of inserted layers to ensure a fair comparison.
>
> ---
>
> ***Other.***
>
> > I find it highly confusing that layers are the “transformer layers” and blocks are the “inner matrices”. This is in contrast to the usual nomenclature in the transformer literature, where blocks are the “outer structure”. I would suggest flipping the names altogether.
>
> We agree that using layers and blocks together can be confusing. However, in depth up-scaling works, layers more commonly refer to the outer structure of the model. This usage is consistent with recent LLM literature, where layers are used to denote model depth (e.g., LLaMA, Qwen, and OLMo). We therefore retain the term layers to describe the outer structure. To avoid confusion, we replace blocks with modules when referring to the inner matrices in the revised version.
>
> > Figure 1 is visually appealing, though not clear what the OpT-DeUS actually does, and it is actually rather confusing; in particular, it is not clear what colors stand for. There is a typo “alinged” (bottom right).
>
> We revised Figure 1 to fix the typo and added additional information in the caption to clarify the use of different colors in the revised version.
>
> > Matrices should use bold symbols. Also, Figures should use the same notation as in 3.1 for clarity. Also, in 3.1, h is not introduced.
>
> In the revised version of the paper, we have updated the entire manuscript to use bold symbols for matrices and revised Section 3.1 to address this issue.
>
> > Figure 2 better state that the right part shows two different strategies, and which is which.
>
> We revised Figure 2 to explicitly indicate the two different strategies.
>
> > An algorithm should be self-contained, i.e., it should introduce all symbols and variables. This is not the case, and it is therefore impossible to understand without reading through the text.
>
> We revised Algorithm 1 in the revised version to make it self-contained.
>
> > I would have liked some more information about the Sinkhorn algorithm, i.e., its intuitive idea that it finds a “soft” alignment, rather than a hard one.
>
> In the revised version, we introduced the relevant information in Section 3.2.
>
> > It is not clear if the model used in the experiments is pretrained off-the-shelf or if the authors trained it from scratch before the experiments.
>
> In the revised version, we clarify in Section 5.1 that the base models are pretrained off-the-shelf.
>
> > Typo “specialzed” in Section 5.4, Typo “ingones” in section Training Efficiency.
>
> Thanks for pointing this out. We corrected these typos in the revised version.
>
> > I would have highly appreciated an ablation on the role of the Sinkhorn regularizer in the final result.
>
> We use the same hyperparameter for the Sinkhorn regularizer following previous work [1]. Analyzing its effect is indeed an interesting direction. However, given our current plan for additional experiments, we temporarily treat this as promising future work and will consider it after completing the planned experiments.

---

> > ### Author Response · Authors · 2026-01-27
> >
> > > I think Table 3 is misleading and does not give any insights: obviously, OpT-DeUS and LLaMA Pro preserve performance at initialization, so I do not see how this is informative for the final performance. Finally, I find extrapolation hard to justify.
> >
> > Following prior work [2], we report post-initialization perplexity to assess up-scaling stability for larger models. We agree that training for a few steps on larger models would provide stronger evidence; however, due to limited computational resources, we prioritize additional ablation studies and will conduct few-step training experiments at larger scales if resources and time permit.
> >
> > > I find it uncommon to report training times, as they are so dependent on actual hardware and implementation. This is confirmed by the fact that some models exhibit much higher training time when run on other hardware. Measuring FLOPs would be a better metric, or at the very least, use the same hardware, or find hardware-independent normalization. Overall, I find this section quite trivial, though; I would probably move it to the appendix.
> >
> > Following the convention in prior up-scaling research [2, 3], we report running hours as a measure of efficiency rather than FLOPs. Due to resource constraints, the 1.72B and 11.5B models are trained on different hardware (detailed in Section 5.6), resulting in differences in absolute training time. However, within each setting, model size and hardware are fixed, making running hours a fair comparison.
> >
> > > Analyzing the nature of the alignment is very interesting. However, there are insufficient details on how the experiments are executed. How is the entropy calculated? Does it depend on the Sinkhorn regularizer used for the alignment? Is it somehow dependent on the dimensionality of the output? How do you account for that? I find it quite odd that the metric varies so little before and after training. I am not sure I follow why low entropy suggests a smaller effective rank.
> >
> > We revised the first paragraph of Section 7.4 and the caption of the corresponding table to clarify how the metric is computed. In the second paragraph of Section 7.4, we explain why the metric varies only marginally before and after training, which is expected. In the third paragraph of Section 7.4, we revised the discussion to more clearly explain the interpretation of low entropy.
> >
> > Explaining the correlation with the dimensionality of the output is indeed an interesting direction. However, given our current additional experimental plan, we temporarily treat this as promising future work and will consider it after completing the planned experiments.
> >
> > ---
> >
> > ***References***
> >
> > [1] Moritz Imfeld, et al. "Transformer Fusion with Optimal Transport." The Twelfth International Conference on Learning Representations. 2024.
> >
> > [2] Yifei Yang, et al. "Lesa: Learnable llm layer scaling-up." Proceedings of the 63rd Annual Meeting of the Association for Computational Linguistics (Volume 1: Long Papers). 2025.
> >
> > [3] Chengyue Wu, et al. "LLaMA Pro: Progressive LLaMA with Block Expansion." Proceedings of the 62nd Annual Meeting of the Association for Computational Linguistics (Volume 1: Long Papers).

---

> > > ### Comment · Reviewer_C481 · 2026-02-03
> > >
> > > Thanks for the thorough response. Could you please clarify whether the current state of the paper is final or if you are still awaiting new results to upload? Just to know when I should read and assess the modified PDF and your response. Thanks!

---

> > > > ### Author Response · Authors · 2026-02-09
> > > >
> > > > Thank you very much for your message. The current version is final, and no further updates are expected due to our current computational resource constraints. Please feel free to proceed with your assessment at your convenience.

---

> > > > > ### Comment · Reviewer_C481 · 2026-02-11
> > > > > **Final Comment**
> > > > >
> > > > > Thanks for the response and the updated manuscript.
> > > > >
> > > > > I find that the new results precisely strengthen the doubts on the experimental setup that I noted in the review, and which are largely shared with Reviewer eA9o. In particular, from Tab. 7, it is rather clear that the gains offered by the proposed method are marginal. This applies both in comparison to some of the design choices of the TMF (e.g. Attention output, MLP gates, where the average zero-shot performance is only slightly below the final method, and they even outperform it in perplexity), and most importantly to other naive methods such as Random+Zero-init. This is most concerning: random initialization + zero-init, is only slightly worse in perplexity and average performance, despite being a highly naive method that clearly should be worse than any sophisticated methods such as OpT-DeUS. Furthermore, all margins are small and not averaged over multiple random seeds, making any claims even more unsupported by empirical results. Finally, and as already argued before, hyperparameters (and in particular LR) should be tuned separately on each method to assess the subjective quality of each one, factoring out mis-tuned confounding hyperparameters.
> > > > >
> > > > > Models with >1B parameters are likely beyond the authors' available computational resources to allow for sound, strong, and fair experimental evaluation. The paper would clearly benefit from having instead smaller models, but compared against well-tuned baselines.
> > > > >
> > > > > In conclusion, the experimental setup does not allow to assess the quality of the proposed method against even highly naive baselines. For this reason, I cannot recommend acceptance of the paper.

---

### Review · Reviewer_eA9o · 2026-01-05

**Summary Of Contributions:**

**Summary:**

Scaling large language models gradually by adding progressively adding new parameters is more efficient than training the model end-to-end directly. The authors consider the specific class of "depth-wise" model expansion approaches to this problem where you increase depth progressively. The current techniques copy or average the weights from the existing layers to initialize the new layers for progressive training. However, the authors conjecture that this ignores the permutation invariance of the layers and it must be accounted for. To this end, the authors propose a method based on optimal transport to align the weights to account for the permutation invariance before averaging adjacent layers to initialize newly inserted layers.

**Weaknesses:**

1. Improvements are not substantial and could be within three standard deviations of error for downstream evaluations. The authors should report the sampling error for the evaluations calculated using bootstraping or any other method to meaningfully interpret the results.
2. Section 4 is difficult to follow as it constantly refers to notation and steps from [1] and [2]. There are many other minor issues. For example, Algorithm 1 is referenced before the notations in the algorithm block are defined. The algorithm description should be fully self-contained in the paper and not require going back and forth between cited references. The authors could consider rewriting this section.
3. Given how small the margins of gains/losses are, it's important to ensure that the differences did not arise from the choices of the hyperparameters. Hyperparameters and the process for tuning hasn't been documented thoroughly. It's not clear from Section 5.5 if the hyperparameters were tuned for each method. For example, the scale of the weights can indirectly affect the learning rate, and I wonder if the distribution of the fused weights for Avg-DeUS and OpT-DeUS are substantially different.
4. The models were severely undertrained for the increase in the number of parameters; for example, the authors trained on 1.5B tokens for the continual pre-training runs after adding nearly 3.5B parameters, which gives a really small Chinchilla multiplier of 0.4. This makes the performance even more sensitive to the choice of hyperparameters.
5. This is a very subjective opinion. I'm not convinced why this method actually results in a meaningful initialization. There are so many arbitrary things (e.g., while handling multiple incoming streams to a layer in the TMF or setting $T_{in}$ to identity for $W_o$) that it feels like the fused weights should affectively be as good as a random initialization. Should the transport plans be analyzed/visualized/interpreted to conclusively establish that the algorithm is finding a meaningful alignment?

[1] Sidak Pal Singh and Martin Jaggi. Model Fusion via Optimal Transport.

[2] Moritz Imfeld, Jacopo Graldi, Marco Giordano, Thomas Hofmann, Sotiris Anagnostidis, and Sidak Pal
Singh. Transformer Fusion with Optimal Transport.

**Additional Comments:**

**A minor nitpick:**

The authors are using architectures that have residual streams. The residual connections can couple the feature spaces across residual blocks to incentivize axis alignment in the representations, which would pressure the model to maintain the neuron/feature ordering across layers. Therefore, for the setting of "copying" based new layer initialization methods, the claim that permutation invariance is not accounted for may not be as concerning as suggested in the article. I agree that it's definitely a signifiant issue for fusing weights though.

**Questions:**
1. What does "multifaceted functionalities" in section 4.1 mean?
2. What is the intuition behind trying to merge weights of two adjacent blocks? Why should this help over random initialization?
3. Is there an explanation for the performance correlations with transport matrix entropy?
4. I'm not able to understand what Figure 2 is trying to illustrate. I understand what's being tried to convey based on the caption but I don't understand what the figure is trying to show. What are the two vertical columns of neurons?
5. I'm not sure about the significance of one of the results based on the following line in the abstract: "extensive analysis shows that inserting new layers closer to the top results in higher training efficiency due to shorter back-propagation time". Is this not a trivial statement that doesn't really require any analysis?
6. How are the values in Table 7 calculated? Is it entropy averaged over all layers? Could this correlation with the scale of the weights, say RMSE?
7. Why is Avg-DeUS absent in Table 2?

**Audience:**

Yes

**Audience Explanation:**

The idea is a straightforward extension of progressive scaling with initializing new layers and attempts to resolve a natural question of what could happen if the initializations were done more carefully. There could be at least some individuals who would be interested.

**Broader Impact Concerns:**

N/A.

**Claims And Evidence:**

No

**Claims Explanation:**

The results for the methods have signifiant statistical reliability concerns due to lack of error estimates for their measurements (and it's well-known that many downstream evals generally have standard errors up to several percentage). Furthermore, there are many concerns regarding their experiment design, particularly undertraining and hyperparameter tuning. They're also lacking a few key baselines such as random initialization and copying $f_i$ over to $f_{i+1}$, which are important at establishing that careful initialization does make a difference. Overall, the authors have not conclusively established that their gains are statistically signifiant and causally related to their algorithm.

**Requested Changes:**

**Critical:**

1. Provide error bars for all measurements (using bootstrapped estimates of sampling error).
2. Train for a multiplier of at least 20x or provide evidence that the models have more or less close to convergence and learning has stalled.
3. Provide details on the hyperparameter tuning effort to show that any gains are not from better tuned hyperparameters.

**Additional ablations (if feasible):**
1. An ablation that could help (partially) address weakness 3 and 5 would be to take the aligned weights obtained from OpT-DeUS  and randomly shuffling the weights elementwise. If this match the performance of OpT-DeUS, then it shows that the spatial statistics do not matter and gives more credence to the suspicion of poor LR tuning affecting the results.
2. What about a randomly initialized new layer for baseline?
3. What about copying $f_i$ over to $f_i'$ with no fusion as a baseline?

**Strengthening the work:**
1. Establish that careful initializations are important and that misalignment is a real problem.
3. Provide visualizations or an analysis of the transport plans (serves as direct causal evidence rather than relying on evals).

---

> ### Author Response · Authors · 2026-01-27
> **Thanks for your thorough review.**
>
> Thanks for your thorough review. Below, let us provide point-to-point responses.
>
> ***Weaknesses***
>
> > Improvements are not substantial and could be within three standard deviations of error for downstream evaluations. The authors should report the sampling error for the evaluations calculated using bootstrapping or any other method to meaningfully interpret the results.
>
> We acknowledge the concern regarding evaluation variance. According to our Additional Experimental Plan, ablation studies are prioritized and are currently being completed. Once these are finished, we will proceed to report the standard error estimated by the lm-evaluation-harness and include evaluation results across multiple random seeds to quantify sampling variability.
>
> > Section 4 is difficult to follow as it constantly refers to notation and steps from [1] and [2]. There are many other minor issues. For example, Algorithm 1 is referenced before the notations in the algorithm block are defined. The algorithm description should be fully self-contained in the paper and not require going back and forth between cited references. The authors could consider rewriting this section.
>
> We agree that Section 4 currently relies too heavily on external references, which makes it difficult to follow. In the revised version of our paper, we rewrite Section 4 and Algorithm 1 to ensure they are self-contained as well as to improve overall clarity and readability.
>
> > Given how small the margins of gains/losses are, it's important to ensure that the differences did not arise from the choices of the hyperparameters. Hyperparameters and the process for tuning hasn't been documented thoroughly. It's not clear from Section 5.5 if the hyperparameters were tuned for each method. For example, the scale of the weights can indirectly affect the learning rate, and I wonder if the distribution of the fused weights for Avg-DeUS and OpT-DeUS are substantially different.
>
> We use the training hyperparameters aligning with the configuration in LESA [1]. Similarly, for all up-scaling baselines, we use the same hyperparameters from the respective original work to up-scale models, varying only the number of inserted layers to ensure a fair comparison.
>
> Due to our Additional Experimental Plan, We will additionally include the analysis of the fused weight distributions for Avg-DeUS and OpT-DeUS to illustrate differences in weight scaling if resources and time permit.
>
>
> > The models were severely undertrained for the increase in the number of parameters; for example, the authors trained on 1.5B tokens for the continual pre-training runs after adding nearly 3.5B parameters, which gives a really small Chinchilla multiplier of 0.4. This makes the performance even more sensitive to the choice of hyperparameters.
>
> Different from prior work [3,4], which focuses on pre-training expanded LLMs, more recent work (including ours and baselines) studies expanded LLMs under fine-tuning or continual pre-training settings with relatively small training budgets compared to full pre-training. We follow prior work [1] and use the same token budget to ensure a fair and consistent comparison.
>
>
> > This is a very subjective opinion. I'm not convinced why this method actually results in a meaningful initialization. There are so many arbitrary things (e.g., while handling multiple incoming streams to a layer in the TMF or setting  to $T_{in}$ identity for $W_O$) that it feels like the fused weights should affectively be as good as a random initialization. Should the transport plans be analyzed/visualized/interpreted to conclusively establish that the algorithm is finding a meaningful alignment?
>
> In the revised version of the paper, we additionally conduct extensive ablation studies in Section 7.3, covering all variants suggested from you in **Additional ablations** section. OpT-DeUS consistently achieves the best performance across all ablation variants, indicating that it provides a meaningful initialization and mitigates neuron permutation mismatch.
> We will further follow your suggestions in **Strengthening the Work** to include visualizations and analyses of the transport plans, if time and resources permit.

---

> > ### Author Response · Authors · 2026-01-27
> >
> > ***Questions***
> >
> > > What does "multifaceted functionalities" in section 4.1 mean?
> >
> > We revise the description in Section 4.1 to make this more illustrative. It refers to, during training, a single neuron may contribute to multiple functions rather than being specialized to a single function.
> >
> > > What is the intuition behind trying to merge weights of two adjacent blocks? Why should this help over random initialization?
> >
> > We revise the second paragraph of Section 4.1 to explain this. Previous depth up-scaling work shows that using information from adjacent blocks to initialize new layers achieves better performance than random initialization.
> >
> > > Is there an explanation for the performance correlations with transport matrix entropy?
> >
> > In the revised version of the paper, we further explain this from the perspective of information theory in the third paragraph of Section 7.4.
> >
> > > I'm not able to understand what Figure 2 is trying to illustrate. I understand what's being tried to convey based on the caption but I don't understand what the figure is trying to show. What are the two vertical columns of neurons?
> >
> > We acknowledge that Figure 2 currently lacks clarity and may hinder understanding. We revise Figure 2 and provide additional information in the caption to make it more illustrative.
> >
> > > I'm not sure about the significance of one of the results based on the following line in the abstract: "extensive analysis shows that inserting new layers closer to the top results in higher training efficiency due to shorter back-propagation time". Is this not a trivial statement that doesn't really require any analysis?
> >
> > While these findings may seem intuitive, prior work did not isolate the effect of insertion position from initialization methods. We decouple these factors to show that training efficiency is governed by insertion position alone. Furthermore, we demonstrate that top-layer insertion yields superior performance, rendering this analysis non-trivial.
> >
> > > How are the values in Table 7 calculated? Is it entropy averaged over all layers? Could this correlation with the scale of the weights, say RMSE?
> >
> > We revise the first paragraph of Section 7.4 and the caption of the corresponding table to clarify how the value is computed and averaged. Explaining the correlation with the scale of the weights is indeed an interesting direction. However, given our current additional experimental plan, we temporarily treat this as promising future work and will consider it after completing the planned experiments.
> >
> > > Why is Avg-DeUS absent in Table 2?
> >
> > Given the extensive ablations we have included, presenting them in Tables 2, 3, and 4 is no longer appropriate. We therefore revise these tables to compare our method only with the baselines. The comparisons involving Avg-DeUS (renamed Average in the revised version) and other ablation variants are now in Section 7.3 and Table 8.
> >
> > ---
> >
> > ***References***
> >
> > [1] Yifei Yang, et al. "Lesa: Learnable llm layer scaling-up." Proceedings of the 63rd Annual Meeting of the Association for Computational Linguistics (Volume 1: Long Papers). 2025.
> >
> > [2] Kazuki Yano, et al. "STEP: Staged Parameter-Efficient Pre-training for Large Language Models." Proceedings of the 2025 Conference of the Nations of the Americas Chapter of the Association for Computational Linguistics: Human Language Technologies (Volume 2: Short Papers).
> >
> > [3] Sheng Shen et al. "Staged Training for Transformer Language Models" Proceedings of the 39th International Conference on Machine Learning, PMLR 162:19893-19908, 2022.
> >
> > [4] Wenyu, Du et al. "Stacking Your Transformers: A Closer Look at Model Growth for Efficient LLM Pre-Training" Advances in Neural Information Processing Systems 2024.

---

> > ### Comment · Reviewer_eA9o · 2026-02-12
> >
> > Were the ablation experiments involve further training or is it just the performance at initialization? The Wiki PPLX is 16.40 for Random. Is this a typo?
> >
> > Overall, Table 7 appears to suggest that the gains depend on the specific task. My overall takeaway is that OpT-DeUS may not be substantially better, if at all better, from the other baselines. The margins are small and the table doesn't have error estimates. The differences in performance between the various methods in the table may not be statistically significant.

---

> > > ### Author Response · Authors · 2026-02-13
> > >
> > > Thank you for the helpful comments. The results in Table 7 are obtained after additional training under the same token budget and training configuration as in the main experiments. The  Wiki PPLX of 16.40 for Random is not a typo.

---

### Review · Reviewer_nyKE · 2026-01-15

**Summary Of Contributions:**

This paper proposes Optimal Transport Depth Up-Scaling (OpT-DeUS), a progressive depth up-scaling method for LLMs that uses Optimal Transport to align and fuse neurons from adjacent base layers when initializing new layers. This addresses the neuron permutation mismatch problem neglected by existing methods that simply copy or average weights. OpT-DeUS achieves better overall performance than baselines across continual pre-training and supervised fine-tuning tasks with various model sizes and domains. The main strength is the principled approach to addressing neuron alignment with extensive empirical validation. A potential weakness is that performance gains over some baselines (e.g., LLaMA PRO) are marginal, and the comparison may not be entirely fair due to different insertion positions of new layers across methods.

**Audience:**

Yes

**Audience Explanation:**

This paper addresses an important gap in model expansion research by introducing a principled approach to layer initialization using Optimal Transport. While existing depth up-scaling methods have relied on simple operations like copying or averaging weights from base layers, this work demonstrates that explicitly addressing neuron permutation mismatch through OT leads to consistent performance improvements. Given the growing interest in efficient scaling methods for LLMs and the broader applicability of the neuron alignment problem, the community of researchers working on model　 expansion and efficient training would find these findings valuable.

**Claims And Evidence:**

Yes

**Claims Explanation:**

The paper provides substantial empirical evidence supporting the effectiveness of using Optimal Transport for new layer initialization. The comprehensive experiments across continual pre-training and supervised fine-tuning demonstrate that OpT-DeUS consistently outperforms baselines on various model sizes and domains. The ablation study comparing OpT-DeUS against Avg-DeUS effectively isolates the contribution of OT-based neuron alignment, and the entropy analysis provides additional insight into why OT-based alignment works better.

However, there are concerns regarding fair comparison and the evaluation of larger-scale stability. Different methods insert new layers at different positions, making it difficult to attribute performance differences solely to the initialization method. For instance, in Table 1, OpT-DeUS-11.5B inserts new layers in the top half of the model, while LLaMA PRO inserts one new layer after every two base layers in an interleaved fashion. This difference in insertion strategy may confound the evaluation of the initialization method itself.
Table 3's evaluation of expansion stability on larger models relies solely on perplexity immediately after initialization. Since OpT-DeUS and LLaMA PRO use zero-initialization for output projections to achieve function preservation, matching the base model's perplexity is guaranteed by design and does not demonstrate robustness during actual training. Although training very large models is computationally expensive, even a few training steps would provide more convincing evidence of stability.

**Requested Changes:**

As noted above, the paper compares methods that insert new layers at different positions (e.g., OpT-DeUS/LESA in the top half vs. LLaMA PRO interleaved throughout the model). This makes it difficult to attribute performance differences solely to initialization methods versus insertion strategy. The paper should either explicitly acknowledge this confounding factor and discuss its implications for interpretation, or conduct controlled experiments where methods use the same insertion positions.
Additionally, Table 3's evaluation of larger-scale stability relies only on post-initialization perplexity. Since OpT-DeUS and LLaMA PRO achieve function preservation by design through zero-initialization, matching the base model's perplexity is guaranteed. While training very large models is expensive, even reporting perplexity after a few training steps would provide more convincing evidence of stability during actual optimization.

---

> ### Author Response · Authors · 2026-01-27
> **Thank you for your valuable feedback.**
>
> Thank you for your valuable feedback. We provide our point-to-point responses in the following.
>
> ***Weaknesses***
>
> > As noted above, the paper compares methods that insert new layers at different positions (e.g., OpT-DeUS/LESA in the top half vs. LLaMA PRO interleaved throughout the model). This makes it difficult to attribute performance differences solely to initialization methods versus insertion strategy. The paper should either explicitly acknowledge this confounding factor and discuss its implications for interpretation, or conduct controlled experiments where methods use the same insertion positions.*
>
> In the revised version of the paper, we further conduct extensive ablation studies in section 7.3 with different initialization strategies (including LLaMA PRO initialization) while using the same insertion position. Under this controlled setting, OpT-DeUS consistently achieves the best performance among all ablation variants, indicating that its gains cannot be solely attributed to insertion position.
>
> > Additionally, Table 3's evaluation of larger-scale stability relies only on post-initialization perplexity. Since OpT-DeUS and LLaMA PRO achieve function preservation by design through zero-initialization, matching the base model's perplexity is guaranteed. While training very large models is expensive, even reporting perplexity after a few training steps would provide more convincing evidence of stability during actual optimization.
>
> Following prior work [1], we report post-initialization perplexity to assess up-scaling stability for larger models. We agree that training a few steps would provide stronger evidence, but due to limited computational resources, we prioritize additional ablation studies and will conduct few-step training experiments if resources and time permit.
>
> ---
>
> ***References***
>
> [1] Yifei Yang, et al. "Lesa: Learnable llm layer scaling-up." Proceedings of the 63rd Annual Meeting of the Association for Computational Linguistics (Volume 1: Long Papers). 2025.

---

### Author Response · Authors · 2026-01-17
**Thank you for your feedback.**

Dear Reviewers,

We sincerely thank you for your constructive and valuable feedback. Below, we first provide a brief, summarized response that addresses common questions and outlines our additional experimental plan. Detailed point-to-point responses to each reviewer’s concerns, along with corresponding revisions, will be provided separately.

---

***Commonly-asked questions***

**On the choices of hyperparameters:** Due to limited computational resources, we do not perform hyperparameter tuning. Instead, we use the training hyperparameters (i.e., include LR) following previous work [1]. For all baselines, we use the same hyperparameters from their respective original works to up-scale models, varying only the number of inserted layers to ensure a fair comparison.

**On the design of TMF:** We do not strictly follow the TMF design in previous work  [2], as architectural advancements in LLMs (i.e., pre-normalization and group-query attention) make it inapplicable. To validate our design choices, we will add additional TMF ablations and revise Section 4 to better justify our TMF design.

**On the number of training tokens:** reviewer applies the Chinchilla scaling law to argue that the number of training tokens is insufficient. However, this is derived for pre-training models from scratch, thus not applicable to our continual training for pre-trained models. In this work, we keep the same training tokens as previous work [1] for consistency.

---

***Additional Experimental Plan***

Due to limited computational resources, we will prioritize and run the additional experiments suggested by the reviewers in the following order, as not all experiments may be completed during the rebuttal period.
- Additional ablations on the choice of TMF.
- Additional ablations verify the effectiveness of OT alignment (i.e., random initialization, random initialization with zero-init, shuffling OT alignment elementwise, copying, average with zero-init).
- Additional evaluation to enhance statistical rigor.
- Additional experiments on larger scale models.
- Additional experiments and analysis on the transport plan.

---

***References***

[1] Yifei Yang, et al. "Lesa: Learnable llm layer scaling-up." Proceedings of the 63rd Annual Meeting of the Association for Computational Linguistics (Volume 1: Long Papers). 2025.

[2] Moritz Imfeld, et al. "Transformer Fusion with Optimal Transport." The Twelfth International Conference on Learning Representations. 2024.

---

> ### Comment · Reviewer_eA9o · 2026-01-17
>
> > On the choices of hyperparameters: Due to limited computational resources, we do not perform hyperparameter tuning. Instead, we use the training hyperparameters (i.e., include LR) following previous work [1]. For all baselines, we use the same hyperparameters from their respective original works to up-scale models, varying only the number of inserted layers to ensure a fair comparison.
>
> The learning rate strongly correlates with the initialization scales. Assuming that the original authors tuned the hyperparameters well, I still do not expect optimization related hyperparameters to transfer.  The learning rate also tends to decay with model size, and the number of inserted layers could alter tuned learning rates too. The nature of this work unfortunately requires careful tuning of the hyperparameters. I'm not sure if the results are meaningful without careful tuning.
>
> Perhaps, at least a couple of simple inexpensive checks such as the LR range test on the baselines, your method, etc. could shed light on whether the learning rates are transferable.
>
> > On the number of training tokens: reviewer applies the Chinchilla scaling law to argue that the number of training tokens is insufficient. However, this is derived for pre-training models from scratch, thus not applicable to our continual training for pre-trained models. In this work, we keep the same training tokens as previous work [1] for consistency.
>
> I agree that the exact scaling law does not hold, and would vary from paper to paper. However, I believe the heuristic that the number of tokens you need is several times the number of new parameters inserted holds in this case (you're adding new layers and you're closer to pretraining regime than you're to fine-tuning). In this particular work, you have a multiplier of 0.5x (w.r.t. newly added parameters) which is alarmingly low and does not instill confidence. I'd have the same concerns with [1].

---

### Decision · Action_Editor_AChs · 2026-02-25

**Recommendation:** Reject

**Additional Comments:**

The paper proposes a new method for progressive depth up-scaling, which is a model expansion approach that adds new layers to a pre-trained LLM without retraining the model from scratch. They focus on post-training model expansion of decoder-only LLMs. The proposed method, Optimal Transport Depth Up-Scaling (OpT-DeUS), inserts new layers in the top half of the base model, interleaved between base layers, and initializes the new layers from adjacent base layers using an optimal transport neuron alignment method, previously used for model merging. Base layers are frozen and only new layers are trained.
Experimental results suggest that OpT-DeUS outperforms baselines, by a small margin, in terms average accuracy on downstream tasks. They also show that inserting new layers in the top half of the base model achieves better training efficiency.

Strengths:
- The paper is well written, particularly after some clarity issues were addressed in the revision, with visually appealing figures that aid understanding.
- Efficiently scaling LLMs is an important topic and progressive depth up-scaling is a promising approach to achieve that.
- The paper proposes a principled approach to address the neuron permutation mismatch limitation of existing depth up-scaling methods

Weaknesses:
- Evaluation results do not include error bars and are not averaged over multiple random seeds.
- The proposed method does not consistently outperform existing baselines; performance varies over different benchmarks. When it does, improvements are marginal, even compared to simple baselines like Random+Zero-init, and may not be statistically significant given the absence of error bars.
- Hyper-parameters were not properly tuned. As reviewers eA9o and C481 pointed out performance can be sensitive to hyper-parameters, particularly to learning rate.
- Up-scaling stability results (Table 3 originally, Table 9 in revision) are not very meaningful as only perplexity at initialization is reported; OpT-DeUS and the baseline LLaMA PRO are guaranteed to preserve perplexity at initialization due to zero-initialization.
- The training budget used might be too small (e.g., 1.5B tokens after adding ~3.5B parameters), it would be good to include results with larger token budgets, or provide evidence that the models are close to convergence.

All reviewers raised key flaws in the experimental setup. Two reviewers recommended to reject on this basis, and one was learning to accept. The authors stated in their response that they plan to address most of these weaknesses, but were not able to do so during the rebuttal period. Unfortunately, with the current experimental setup it is not possible to judge if the proposed method outperforms existing baselines as claimed.

I am thus recommending to reject the paper, with the option of submitting a major revision at a later time.

Requested revisions: Additional/modified experimental evaluation to address the above weaknesses.

Other revisions recommended by reviewers:
- Provide visualizations or an analysis of the transport plans
- Present an ablation on the role of the Sinkhorn regularizer

**Audience:**

Yes

**Audience Explanation:**

Depth up-scaling is a promising approach for efficiently scaling LLMs, which is a very relevant and timely topic. The paper aims to address a significant limitation of existing depth up-scaling methods. The presented results would be of interest to researchers working on efficient LLM training if well supported.

**Claims And Evidence:**

No

**Claims Explanation:**

The experimental setup has several important issues outlined below in weaknesses. Thus experimental results do not support the claim that the proposed method outperforms baselines across various model sizes and diverse tasks.

**Resubmission Of Major Revision:**

The authors may consider submitting a major revision at a later time.